# Online Song Intervention Program to Cope with Work Distress of Remote Dispatched Workers: Music for an Adaptive Environment in the Hyperconnected Era

**DOI:** 10.3390/bs15070869

**Published:** 2025-06-26

**Authors:** Yaming Wei, Hyun Ju Chong

**Affiliations:** 1Research Center for Arts and Health, Xiamen Humanity Hospital, Xiamen 361005, China; yamingmt@gmail.com; 2Music Therapy Department, Ewha Womans University, Seoul 020844, Republic of Korea

**Keywords:** work-related distress, psychological resources, content analysis, song intervention program, contextual support model

## Abstract

With the increasing demands of long-term overseas assignments, workers in isolated environments, such as maritime crews, often experience heightened psychological stress and a lack of accessible emotional support. This study investigates the effectiveness of online song intervention program based on contextual support model in reducing work-related distress and enhancing psychological resilience among the ship crews dispatched for an extensive period for work. Eighteen overseas workers participated in a four-week intervention that included both individual and group sessions, where they engaged with songs to cultivate personal and interpersonal resources. A deductive content analysis following the intervention revealed 3 main categories, 6 generic categories, and 14 subcategories. The three main categories identified were relationships, autonomy, and mood regulation. The relationships category encompassed support systems and bonding, focusing on empathy, consolation, positive perspective, vicarious empowerment, trust, and changes of perspective. Autonomy involved fostering a sense of control and fulfillment through determination, anticipation, motivation, and achievement. Mood regulation was divided into grounding and emotional resolution, which included containment, sedation, externalization, and ventilation. The findings highlight that song lyrics offer valuable insights for developing resources aimed at mood regulation, social support, and self-efficacy, helping to alleviate work-related stress during dispatch periods. Songs also foster a sense of control, competence, and relational connectedness, with mood regulation emerging as a key feature of their emotional impact. These results suggest that incorporating songs with lyrics focused on personal and interpersonal resources could be an effective strategy to support remotely dispatched workers. Furthermore, this approach appears to be a viable and scalable solution for online programs.

## 1. Introduction

In the era of economic globalization, multinational companies frequently dispatch employees overseas for extended periods to meet their strategic needs. Dispatched employees are typically engaged in international assignments, outsourcing projects, or temporary personnel arrangements. These employees include company personnel sent to work in another country, usually as part of a project, contract, or corporate expansion, as well as those hired by agencies and assigned to work at client companies for a certain period. These workers face the dual challenge of adapting to unfamiliar cultures, languages, and work environments while maintaining productivity, which poses unique psychological challenges and emotional health issue ([8]; [31]).

Emotional health, encompassing emotional stability, positivity, and stress management, is critical to physical and mental well-being. Impaired emotional health can lead to negative outcomes such as anxiety, decreased sleep quality, and weakened immunity. These stressors are particularly severe for overseas workers lacking psychological resources in unfamiliar and often isolating environments ([11]; [20]). Given these specific challenges, traditional psychological support structures are often unavailable or insufficient for long-term and personalized inner resource support. While traditional methods such as counseling, cultural training, and communication support have improved workers’ mental health and job satisfaction to some extent ([24]). dispatched workers often require more time for verbal interaction and sustained support. Considering these significant demands, innovative and personalized approaches are needed to enhance emotional well-being and resilience.

Songs uniquely combine music and language, conveying meaningful messages and emotional narratives while promoting emotional expression, social interaction, and self-reflection ([18]). In music therapy, songs engage listeners through comprehensible lyrics that address emotional and psychological states and reflect the songwriter’s thoughts and feelings ([5]). This process facilitates direct emotional engagement, making song intervention an effective tool for emotional regulation and a valuable resource for individuals in isolated environments. By selecting song materials based on a contextual support framework, therapy can address a range of life issues, from repressed emotions to fostering autonomy, relatedness, and competence. These qualities assist in stress management and enhance psychological resilience and self-efficacy, defined as leveraging internal strengths and external support to manage stress and stabilize emotions ([3]; [32]). As a fusion of lyrics and musical content, songs provide cognitive and emotional stimulation that supports resilience in demanding environments.

Extensive research highlights the efficacy of music in supporting emotional regulation. Music enables individuals to access and process complex emotions, boosting self-esteem and fostering social recognition ([7]). Song lyrics in music can resonate with the listener’s experiences, creating avenues for self-reflection and stress relief ([9]). Song intervention has demonstrated significant efficacy in populations experiencing deficient inner resources due to social isolation, neglected adolescents or individuals recovering from trauma ([12]; [17]). These findings support the potential of song intervention as an emotional support tool for socially isolated workers, particularly in alleviating loneliness and enhancing social connectivity.

The contextual support model is a psychological framework describing how environmental and situational factors influence motivation and behavior. The theory outlines three elements of coping and stress management: autonomy, competence, and relatedness ([25], [26]). In this theoretical framework, autonomy refers to the freedom to make decisions, relatedness to having secure and meaningful relationships both internally and externally, and competence to a sense of control. These three elements contribute to coping, which describes how individuals regulate their behaviors, emotions, and motivational orientations in psychologically distressing situations ([29]).

One of the many limitations of traditional psychotherapy is the restricted access to experts who can communicate in the individual’s native language, coupled with the time and location constraints of therapy sessions. Even when experts are proficient in the native language, scheduling therapy outside of working hours remains challenging. In the era of hyper-connectivity, music and song materials can transcend physical boundaries, offering immediate access when need ([2]). The benefits of hyper-connectivity can be adapted into therapeutic programs to meet the specific needs of workers in remote areas who cannot access therapy experts. Many employees are dispatched to other countries for extended periods, with some situations being particularly intense, such as employees in the shipping industry. Song intervention provided via online connections can serve as a psychological resource and become a significant program for employers. For example, it can benefit workers on ships undertaking prolonged voyages in multiple foreign regions.

This study aimed to develop and evaluate an online song intervention program based on the contextual support model to help ship crews alleviate work-related distress and enhance their psychological adaptation to challenging working environments. By utilizing songs therapeutically, the intervention sought to facilitate access to latent emotional and psychological resources, thereby fostering resilience in the face of occupational stress. The study addressed the following research questions:Does the online song therapy intervention help participants better cope with work-related stressors?How does the intervention foster psychological resilience through relationship building, autonomy, and emotional regulation.What key changes in participants’ psychological processes emerged during and after the intervention?How can participants use the song resource in their work environment?

## 2. Method

### 2.1. Participants

The participants’ company was a Chinese enterprise, headquartered in China, and a world-class overseas shipping company with workers stationed globally. Participants were recruited from the crew members of this shipping company who had been dispatched abroad for at least six months to one year, with an average continuous working period on the ship of no less than three months. The inclusion criteria for participation in the study were as follows: (1) aged between 20 and 60 years; (2) dispatched abroad for more than one year; and (3) without any hearing or speech impairments. Recruitment was conducted via online platforms, and participation was entirely voluntary. Importantly, all participants explicitly expressed a perceived need for psychological and emotional support and provided informed consent to participate in the intervention program after receiving detailed information regarding its objectives, structure, and anticipated benefits. A total of 18 Chinese dispatched workers consented to participate and signed confidentiality agreements prior to the commencement of the study (Table 1).

### 2.2. Procedure

The study followed a four-stage process to design, implement, and evaluate the intervention (Figure 1). In the first stage, participants were assessed for readiness and psychological mindedness through an intake questionnaire, which also identified their preferred and personally meaningful songs. The questionnaire was mainly open-ended and scored. These songs were compiled to establish a dedicated music library for the intervention. The assessment provided essential information regarding participants’ emotional resources, interests, and personal circumstances, enabling the therapist to tailor the intervention accordingly.

The second stage comprised a four-week intervention involving both individual and group sessions conducted once per week. The details of the intervention formulation and implementation, following [27]’s ([27]) Reporting Guidelines for Music-Based Interventions (RG-MBIs). Individual sessions, each lasting 45 to 60 min, focused on participants’ personally meaningful song selections, aiming to address their unique emotional needs and facilitate resource development. Conversely, group sessions, lasting 60 to 90 min, utilized songs frequently selected across participants to promote shared experiences, enhance group cohesion, and foster mutual support. The selection of songs was informed by frequency analysis of preference surveys and assessment outcomes.

In the third stage, the intervention was formally conducted by a doctoral candidate in music therapy, who also served as the primary researcher. The therapist was supervised by a professor with over 20 years of clinical experience. All intervention plans underwent pre-review to ensure accuracy and clinical appropriateness. Music was delivered via recorded tracks, played through speakers in group sessions and through headphones in individual sessions. Lyric sheets and questionnaires for self-reflection were provided to facilitate deeper engagement with the music content. Group sessions typically included 5 to 10 participants from the same ship, fostering an atmosphere of collective sharing and support, while individual sessions ensured a personalized therapeutic environment.

The final stage involved qualitative content analysis to examine how participants engaged with the song materials and how psychological and emotional resources were activated during the intervention. Treatment fidelity was maintained through strict adherence to the structured intervention protocol and continuous monitoring throughout the process. The intervention was conducted in a controlled environment to minimize distractions and optimize participant comfort. All procedures and ethical considerations were reviewed and approved by the Institutional Review Board of Fuzhou University (FZU-PSY2024-0165, 21 March 2024).

#### 2.2.1. Intake for Participants Psychological Needs and Therapy Readiness

To ensure the effective development and implementation of the song intervention, this study incorporated intake questionnaires to examine each participant’s readiness to work on their well-being and psychological mindedness in using songs as a psychological resource. The questionnaire consisted of multiple-choice items and one open-ended question inquiring about any additional issues to be addressed. The intake questionnaires collected essential information regarding participants’ current issues and their expectations for therapy. The questionnaire included six items, which were reviewed and validated by three music therapy professionals to ensure content validity. The review ensured that the questions appropriately addressed the intended concepts related to readiness and psychological mindedness, as shown in Table 2. In addition to this intake, information about musical preferences and musical background was collected to help formulate appropriate song materials for the intervention.

#### 2.2.2. Development of Song Intervention Program

The song intervention program was developed based on the contextual support model. This theoretical framework posits that environment supporting these core needs foster motivation and psychological well-being. Within this context, the intentional use of songs with meaningful lyrical content served as a mechanism to activate intrinsic emotional resources, promote self-reflection, and foster interpersonal connections. Specifically, lyrical themes were selected to evoke positive memories, enhance emotional resilience, and facilitate adaptive coping strategies in response to the stressors associated with working in overseas environments as shown in Figure 2.

The intervention consisted of a structured combination of both individual and group music therapy sessions delivered over a four-week period. Each week included one individual session and one group session. A total of eight songs were selected for the intervention (Table 3). The selection of songs followed clearly defined criteria to ensure both therapeutic relevance and cultural appropriateness (Appendix A). Detailed information about the specific songs selected for use in the intervention, including song titles, artists, thematic relevance, and musical characteristics, are provided in Appendix B.

The individual sessions were systematically structured to engage with intrinsic motivation for participation and interest in music, with the primary aim of facilitating emotional identification and activating personal resources. In the first session, participants were encouraged to identify any positive associations related to memories or episodes, thereby fostering autonomous engagement with music and supporting the development of initial interest. The second session emphasized the identification of songs that reflect ‘here & now,’ guiding participants to select and reflect on lyrics that depict their present feelings and thoughts. Building on this process, the third session focused on identifying personal ‘power’ songs and affirming their ownership, thereby enhancing participants’ recognition of personal resources for intrinsic strength. Furthermore, participants were encouraged to share songs that promote proactive perspectives, facilitating the activation of self-efficacy. In the final session, participants were guided to write empowering personal lyrics to a familiar tune, thereby creating an adaptive environment for sustained motivation and long-term emotional well-being.

The group sessions were designed to emphasize interpersonal relationships and to facilitate the development of shared emotional experiences through collaborative engagement with music. The first session encouraged participants to share familiar songs with the group and stories related to them, thereby promoting group belonging and strengthening group identity. In the second session, participants were invited to identify common themes in lyrics shared by the group, using songs about life issues, difficulties, and challenges to foster emotional bonds and trust among members. The third session further deepened interpersonal engagement by encouraging participants to share songs that promote proactive perspectives on current challenges, facilitating the affirmation of shared strength within the group. The final session guided participants to write group lyrics about coping strategies for current situation to the shared tune, thereby consolidating the group as a collective support system. Through this process, the group environment served as a platform to support determination, perseverance, and a positive outlook, promoting mutual resilience resource.

It is important to note that the objectives of each session are not expected to be achieved within a single session, instead progressively cultivated throughout the course of the entire intervention. Ultimately, each session would contribute incrementally to the development of both personal and collective resources. Each session was recorded and transcribed prior to the following session. The transcribed qualitative data were examined to formulate each intervention. The implemented information is shown in Table 4 to report details of the formulation using [27] ([27])’s Report Guideline for Music-based Interventions (RG-MBIs) as shown.

#### 2.2.3. Participation Format for the Program

The intervention was conducted online using digital devices, as the participants were stationed on a ship during the study period. Individual sessions were held in the participants’ private cabins, where they used either mobile devices or laptops depending on their individual circumstances. Group sessions were conducted in the ship’s activity room, which was equipped with a large monitor sitting in a semi-circle. Individual sessions were carried out in a private closed environment. As work-related issues evolve from interpersonal dynamics, the individual sessions addressed them with confidentiality.

#### 2.2.4. Data Collection and Analysis

Data were collected from participant narratives during both individual and group sessions, as well as from post-intervention interviews conducted online after the completion of the program. All sessions and interviews were audio-recorded and transcribed verbatim to ensure accuracy and completeness. The transcribed data were securely stored on a dedicated hard drive accessible only to the principal researcher, thereby maintaining confidentiality throughout the research process. Following each session, participants provided oral responses to a post-session questionnaire, focusing on their emotional reactions to the selected songs and any perceived psychological benefits. After completing the four-week intervention, individual post-intervention interviews were conducted to further explore participants’ reflections on the overall experience, emotional and psychological changes, and their views on the applicability of the intervention in similar contexts.

The transcribed data were analyzed using deductive content analysis, guided by predetermined theoretical constructs related to psychological resources. This approach provided a systematic framework for organizing and interpreting the data, ensuring coherence and grounding the findings in established theory. The transcripts were coded and categorized according to key concepts linked to the therapeutic functions of songs. Recurring patterns and themes were then examined in relation to the theoretical framework to confirm their consistency and enhance the analytical depth of the study. To enhance data credibility, key findings were confirmed with participants during the interview process. This approach yielded a comprehensive understanding of how the song lyrics reflected and evoked psychological resources at both personal and interpersonal levels. The post-intervention interview questions were as follows:

## 3. Results

The categorization of the word units and groupings yielded 14 subcategories in which participants’ verbal statements were categorized according to the themes and key words related to beneficial aspects of one song. Each subcategory was operationally defined, and sample statements are shown in Table 5. The coding of the displayed parts is marked on the right side. Here are some examples: P1-1 indicates verbal statement provided by Participant 1 verbalized in their first individual session. Likewise, G2-2 means verbal statement provided by Participant 2, in their group session. INT4 means verbal statement provided by Participant 4 during their interview session (Table 6).

Following the grouping of individual verbal statements into subcategories, generic categories were identified based on thematic similarities. A total of six generic categories emerged: support system, bonding, sense of control, sense of fulfillment, grounding, and emotional resolution. These six generic categories collectively captured the essence of three overarching main themes: relationship, autonomy, and emotion regulation. Relationship referred to resources related to interpersonal connections, attitudes toward others, and openness in social interactions. Autonomy encompassed intrapersonal domains, such as personal perspectives, belief systems, and self-efficacy. Emotion regulation described the psychological processes through which songs facilitated the expression or resolution of excessive or unresolved negative emotions (Table 7).

## 4. Discussion

This study provides practical meaning of using songs for therapeutic purpose to enhance intrinsic psychological resources and reduce stress among overseas dispatched workers. The intervention aimed to activate both internal and external resources, supporting participants’ adjustment to the challenges of working in a remote, distanced environment. Based on the experiences shared by participants, several significant theoretical and practical implications emerged.

The results indicate that song lyrics can serve as a potent psychological resource in the dispatched work environment. Participants often engaged with lyrics that resonated deeply with their immediate personal and professional concerns, finding messages of personal relevance within the songs. This finding aligns with the theory of musical intelligence, which suggests that individuals have an innate capacity to use music as a tool for emotional self-care and psychological insight ([21]). By connecting with lyrics, participants could express, which promoted emotional processing and self-awareness.

Moreover, the songs selected based on the key concepts of the contextual support model facilitated participants’ exploration of various aspects of their personal and professional lives, including areas related to self-efficacy. For example, one participant shared, *“I decided to do one small thing every day starting from today, even if it is to read a page of a book.”* (S1-8), illustrating how lyrical content activated a sense of agency and personal goal setting. Another participant noted, *“Sometimes it is much easier to give up than to persist. If I persist, I hope I see gains.”* (P10-3), reflecting the link between music engagement and self-efficacy beliefs. The group dynamics fostered a supportive environment that enabled participants to share their interpretations and personal narratives connected to the songs. For instance, one participant described, *“I got along very superficially before, but now we have become more comfortable after having deeper communication.”* (G3-8). This collective engagement contributed to the activation of interpersonal resources, promoting empathy, mutual support, and a sense of community. Participants described experiencing a sense of empowerment from the shared lyrical content and reported that listening to others’ experiences encouraged their own self-reflection: *“In the past, I relied on my own internal support. Through the conversation just now, I realized the importance of team support.”* (G2-9). These findings indicate that songs can serve as a medium for accessing internal psychological resources while simultaneously fostering social connectedness—factors that may support emotional resilience in isolated work environments ([1]; [6]; [22]; [30]).

The study also found that song lyrics with positive or nostalgic themes helped participants access memories that reinforced their motivation and optimism. Reflecting on uplifting lyrics allowed workers to reconnect with joyful past experiences, which improved their mood and provided a temporary escape from the challenges of a foreign work environment. For example, *“Like the lyrics, I’m hopeful that things will work out for the best. I believe good things are on the horizon.”* (S2-8), and *“A dream I had forgotten about for a long time was suddenly ignited, and I wanted to continue to realize it.”* (G2-9) both illustrate the activation of positive memory recall and forward-oriented optimism. This process, referred to as positive episodic memory recall, has been shown to enhance mood regulation by evoking meaningful, life-affirming memories ([10]; [13]; [28]). By inducing emotional recall, song interventions may boost intrinsic motivation and resilience among workers facing prolonged isolation or cultural adaptation challenges. Tailoring playlists or music interventions to include songs that resonate personally with workers may thus serve as a strategic approach for enhancing well-being. This aligns with existing research on the therapeutic functions of lyrics in building emotional resilience and providing comfort during stressful periods ([4]; [14]; [15]; [23]).

The findings also highlight the unique role of songs as a medium for both verbal and nonverbal expression of emotions. Many participants noted that the music enabled them to express feelings that were otherwise difficult to articulate. For example, one participant expressed, *“I want to confess my inner thoughts to the therapist.”* (P9-3), while another reflected, *“When I loudly sang the words, I filled in by myself, which gives me a great sense of accomplishment and satisfaction.”* (P11-3). Encouraging participants to sing or listen to songs reflecting their emotions created an emotional outlet, reducing the psychological burden of unexpressed stress. *“The melody relaxes my body easily and makes my whole heart soft.”* (P4-3), noted another participant, underscoring how embodied musical engagement facilitated emotional regulation. This supports the theoretical concept of music as a mode of emotional release, which posits that music can provide an avenue for emotional catharsis and self-expression ([19]) By offering a channel for nonverbal emotional expression, song intervention allows individuals to externalize complex emotions, fostering emotional balance and reducing work-related distress.

Lastly, song lyrics helped participants project and regulate repressed emotions, as music can evoke strong emotional responses. For many participants, externalizing emotions through lyrics allowed them to vicariously articulate feelings and thoughts they might otherwise have suppressed. *“I learned how to control bad thoughts through music, which I think is a great gain.”* (P6-3) and *“I shifted my attention to music and stopped thinking about those bad things.”* (P2-3) both exemplify this process of emotional projection. This enabled participants to “ventilate” stress and work-related concerns, contributing to mood regulation and mental relief ([33]). The projection of emotions through music echoes the psychological theory of externalization, where individuals find relief by expressing emotions through a separate medium vicariously. Song lyrics, by voicing these emotions through words, help participants confront and manage their internal struggles, promoting psychological resilience in the face of work challenges. Moreover, as examined in the literature review, another salient feature reported by the participants was the emotion regulation aspect of music. *“I don’t worry about so many things that will happen in the future and focus on the present.”* (P1-3), said one participant, while another shared, *“The familiar melody makes me feel the sense of peace.”* (INT7). These findings support previous research indicating that music enables individuals to access and process complex emotions, which can boost self-esteem and foster social acceptance ([7]; [16]). These results further reinforce the therapeutic role of music as a medium for emotion regulation, particularly in challenging work environments.

### Limitations and Suggestions for Future Directions

This study developed a short-term, structured song intervention program aimed at helping participants build psychological resources to improve their work performance and adaptation to confined environments. However, several limitations should be acknowledged. First, no follow-up assessments were conducted after the completion of the program, which limited the ability to evaluate the long-term effectiveness of the intervention. Incorporating follow-up evaluations in future studies would provide valuable insights into the sustainability of the intervention’s psychological benefits.

Second, the study sample consisted exclusively of male participants due to the occupational characteristics of the target population. This gender imbalance limits the generalizability of the findings. Future research should aim to include a more diverse participant pool to enhance the applicability of the results across broader populations.

Finally, although the group setting provided opportunities for shared emotional experiences and mutual support, the unique social dynamics of the participants—who lived and worked together in confined environments for extended periods—may have inhibited their willingness to fully disclose personal emotions or vulnerabilities during group sessions. Future interventions should therefore consider strategies to manage group dynamics more effectively, ensuring a balance between fostering group cohesion and safeguarding individual emotional privacy.

In addition, while this study focused on a specific occupational context, future research could explore the adaptability and effectiveness of the song intervention in other work environments, particularly in remote or isolated occupational settings. Expanding the scope to different professional fields would contribute to understanding the versatility and broader applicability of the intervention.

Furthermore, with the continuous advancement of digital technology, future studies may consider integrating technology-based platforms, such as mobile applications or virtual reality environments, for delivering song interventions. This approach holds potential for increasing accessibility, personalizing the intervention process, and enhancing user engagement, thereby expanding the impact of song-based therapeutic programs across various contexts.

## Figures and Tables

**Figure 1 behavsci-15-00869-f001:**
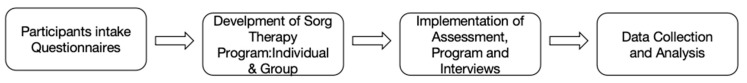
Procedure in Four Stages.

**Figure 2 behavsci-15-00869-f002:**
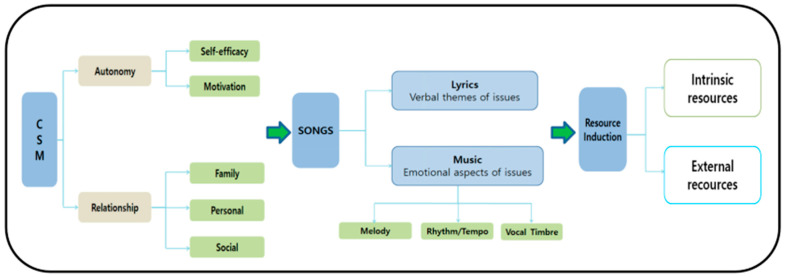
Song Intervention Program Development under Contextual support Model.

**Table 1 behavsci-15-00869-t001:** Demographic Information of Participants.

(*N* = 18)		
Variable	*n*	%
Gender		
Male	18	100
Female	0	0
Age (years)		
20–29	7	38.9
30–39	7	38.9
40–49	3	16.7
50 and above	1	5.6
Participant type department		
Technology	7	38.9
Shipping Crew (Cargo loading)	7	38.9
Management	4	22.2
Length of working on the ship (years)		
4–6	9	50
7–9	8	44.4
10 and above	1	5.6
Length of time you last worked on board a ship (months)		
3–4	7	38.9
5–6	9	50
6 and above	2	11.1
Prior experience of therapy		
Yes	2	11.1
No	16	88.9

**Table 2 behavsci-15-00869-t002:** Intake Questionnaires for Song Intervention Program.

Section	Questionnaires
Readiness	- Willingness to understand oneself and others - Motivation to participate and degree of expectation - Openness for new perspectives and changes
Psychological mindedness	- Ability to self-reflect and identify - Concept of one’s unconsciousness and psychological minds - Willingness to explore insights for self-care

**Table 3 behavsci-15-00869-t003:** Individual and Group Session Goals and Content.

Session No.	Session Objectives	Session Format		Song Lyrics for Resource
		Individual	Group	
1	Engage with intrinsic motivation for participation and interest in music	Identify any positive associations related to memories or episodes	Share familiar songs with the group and stories related to them	Songs about personal memories, family issues, hometowns, etc.
2	Identify songs that reflect ‘here & now’	Select and reflect on lyrics that depict the present emotions and thoughts	Identify common emotional themes in lyrics shared by the group.	Songs about emotions and thoughts related to life issues, loneliness, homesick, challenges, etc.
3	Identify music resources for personal (intrinsic) and the group (shared) strength.	Identify personal ‘power’ songs and affirm the ownership	Share songs that promote proactive perspectives on current challenges	Songs expressing commitment, inner strength, self-efficiency, etc.
4	Use the song resources to create an adaptive environment for sustained motivation	Write empowering personal lyrics to a familiar tune.	Write a group lyrics to cope with current situation to the shared tune.	Song writing for determination, perseverance, and a positive outlook.

**Table 4 behavsci-15-00869-t004:** Report Guideline for Music-based Interventions.

Components	Sub-Components	Descriptions/Rationale
Intervention Rationale	Theoretical framework	Based on Contextual Support Model concepts: Motivation, self-efficacy, sense of autonomy, psychological resources, interpersonal relationships, etc.
Music selection	Selection Rationale	Selected based on the participant’s demographic information, reported familiarity of the song repertoire, session objectives, etc. Lyrics of the selected songs were analyzed for its feasibility and coherence of the session objectives
Music Delivery Method	During Session	Played via on-line devices
	Outside of Session	Personal music playing devices
Materials	Lyrics	Sheet music with lyrics of selected songs are provided
	Pen/papers for reflection writing	Writing materials for personal song writing
Intervention Strategies	Music listening	Listening to the general mood of the music
	Song discussion	Discussion of lyrics with target themes and meaning
Interventionist	Primary therapist	Credentialed music therapist
Intervention format	Individual and group sessions are provided alternatively each week	Individual and group sessions were provided alternatively to address issues that were personal and relational. In the individual sessions, various personal issues were explored whereas in group sessions, interpersonal issues were shared.
Setting	Individual sessions	Participant’s choice of private area such as cabin, where they could use laptop and speakers/microphones.
	Group sessions	Activity room equipped with big screen and speakers for online program implementation
Intervention Delivery Schedule	Individual sessions	Once a week for 60 min
	Group Session	Once a week 90 min
Treatment Fidelity	Session report	Each session was recorded and transcribed to collect their verbal statements regarding the song therapy
	Lyric materials	Participant’s substituted lyrics are collected to analyze personal, emotional and social resources expressed through lyrics.
	Post-session Interview	-Session questionnaire: After every session, participants were asked about the lyrics and their current state. -Post-program interview: Individual interviews were conducted after the four-week overall intervention, lasting 30 min per participant.

**Table 5 behavsci-15-00869-t005:** Interview questions.

	Categories	Sample Questions
I	Associations or memories related to the lyric	- What part of the lyrics comes to your mind? Any part of the lyrics that reflect your thoughts or feelings? - What memories do the lyrics bring up for you? - Could you share memories or associations?
II	Messages from the lyrics	- Were you able to identify what the song lyrics suggest? - Were you able to relate to the singer or empathize with the singer’s experience? - Anything related to self, others or environments?
III	Integrate the message of lyrics into daily life	- When do you think you can use imagery or messages for yourself in everyday life? - How can they help you with your work attitude?
IV	Owning the song as coping strategy	- How can you use this song for other parts of your life, such as relationships? - What other benefits does this song give you?

**Table 6 behavsci-15-00869-t006:** Operational definitions of subcategories and sample statements.

Generic Category	Operational Definition	Sample Statement
Empathy	- Includes statements to understand and relate to each other’s emotions. - Validates feelings and strengthens interpersonal connections.	- I can feel the emotion in the singer’s voice. (P3-1) - I’ve been through similar situations before, and I know how you’re feeling right now. (G4-12) - I felt the joy you described in lyrics. (G2-8) - The rise and fall of the melody matched my emotions. (INT1)
Consolation	- Any kind of solace after disappointment, pain, or sadness. - Any positive support to help alleviate emotional pain.	- I am dealing with this struggle on my own, but music gives me comfort. (INT3) - The steady music rhythm gives me a very stable feeling. I felt someone gently patting my shoulder to comfort me. (P12-1) - Music filled the void and I felt a sense of comfort like never before. (S5-3) - The “it’s okay” in the lyrics comforts me. (INT2)
Positive Perspective	- Focus on strengths and achievements, an optimistic perspective. - Perspectives regarding future outlook	- I believe that if we keep a positive attitude and work hard, as the lyrics say, “everything will be alright.” (G1-3) - His courage and perseverance inspire me. (G3-9) - His views touched me a lot, and I decided to make changes. (G2-5) - I hope that, as he says, everything that follows is as simple as breakfast. (G4-7)
Vicarious empowerment	- Feeling empowered through the success or strength of others. - Motivate yourself through the experiences of others on the team, making yourself more confident and motivated to pursue your goals.	- The lyrics “Push open the door to the world” gave me a lot of encouragement and courage. I want to be the one who opens the door to see the world. (S2-6) - This song conveys positive energy, and I believe I can do it. (S3-5) - In the past, I relied on my own internal support. Through the conversation just now, I realized the importance of team support. (G2-9) - I am very small in the entire ocean, but that does not mean that we are unimportant. We are also a very important part of this world. (INT8)
Trust	- Trust that the other person can listen and understand you and trust each other. - Having an open mind to share.	- I want to confess my inner thoughts to the therapist. (P9-3) - I got along very superficially before, but now we have become more comfortable after having deeper communication. (G3-8) - I put my “life” in your hands. (P7-3) - People you spend time with every day are more trustworthy. (INT5)
Change of Attitude	- Accepting a different attitude toward a situation or an event than before.	- I’ve come to understand the importance of refraining from defining the quality of my current work situation solely through a worldly lens. (P1-4) - Presently, I believe that having a fulfilling job, stable income, and a happy family equates to success. (P6-3) - It’s essential not to solely focus on the outcome’s quality; the process itself holds equal significance. (P7-2) - I would change sides and enjoy my time instead of wasting it anxiously. (INT2)
Determination	- Reconnecting with the resolution and previous intentions. - Feeling new strength for pursuing goals.	- The lyrics remind me not to hesitate anymore and to embrace everything I have now. (P5-2) - Sometimes it is much easier to give up than to persist. If I persist, I hope I see gains. (P10-3) - I decided to do one small thing every day starting from today, even if it is to read a page of a book. (S1-8) - Overcoming the boredom of work is a very important thing for me now (S3-7)
Anticipation	- Positive mind for what is to come. - Feeling excited about the achievement.	- Like the lyrics I’m hopeful that things will work out for the best. I believe good things are on the horizon. (S2-8) - Music gives me feeling to travel, I’m really looking forward to the upcoming vacation. It’s going to be an amazing experience! (P5-2) - I’m optimistic about what the future holds. I believe there are great things in store for me. (P11-3) - The rising sense of melody gives me hope like the sunrise. (INT2)
Motivation	- Intrinsic motivation and internal desire to move forward.	- The fast pace of the music is very immersive and makes me full of motivation to do things. (P10-3) - The lyrics remind me of my high-spirited self at that time and inspire my lazy self now. (INT9) - A dream I had forgotten about for a long time was suddenly ignited, and I wanted to continue to realize it. (G2-9) - Even when things get tough, keep going. Don’t be afraid to move forward. After listening to the music, I had such an impulse. (INT12)
Achievement	- Sense of achievement experienced through music that can be applied to other areas. - Sharing positive feelings about what was achieved or accomplished.	- When I loudly singing the words, I filled in by myself gives me a great sense of accomplishment and satisfaction. (P11-3) - I feel like I can see my inner thoughts more and more clearly. This is an improvement for me. (S2-9) - Sure enough, the road to overcoming difficulties is full of challenges and obstacles, but it also makes success even sweeter. (S4-11)
Containment	- Any emotional skills not being overwhelmed by negative thoughts or feelings. - Ability to emotionally regulate.	- I learned how to control bad thoughts through music, which I think is a great gain. (P6-3) - I shifted my attention to music and stop thinking about those bad things. (P2-3) - It’s not a bad thing to stop occasionally, it allows us to rethink and start again. (INT3) - I don’t worry about so many things that will happen in the future and focus on the present. (P1-3)
Sedative Centering	- Soothing or calming of emotional arousal to promote feelings of calm and stability.	- The familiar melody makes me feel the sense of peace. (INT7) - Music gives me some space and makes me feel like I have a higher oxygen level in this cabin. (S4-7) - The ascending melody gives me hope like the sunrise. (INT2)
Externalization	- Expressing emotions, thoughts, or experiences outside of themselves.	- I really want to sing loudly, but I can’t seem to do it. (P2-4) - The melody relaxes my body easily and makes my whole heart soft. (P4-3) - I think it would be a good idea to dance to this beat or listen to it during my work out tomorrow. (P9-3)
Ventilation	- Letting out or releasing repressed stress reaching emotional relief.	- Heavy beats make me feel comfortable and release my stress. I like rock music. (INT10) - The rhythm of this music hit me like a punch in the chest, making me feel like I had more room to breathe. (S3-4) - Music opens a new world. I feel like the world is not just as big as a ship, and I can breathe in oxygen. (S4-12)

**Table 7 behavsci-15-00869-t007:** Deductive content analysis through abstraction process.

Subcategory	Generic Category	Main Category
Empathy	Support system	Relationship
Consolation		
Positive perspective		
Vicarious empowerment	Bonding	
Trust		
Changes in perspective		
Determination	Sense of control	Autonomy
Anticipation		
Motivation	Sense of fulfillment	Competency
Achievement		
Containment	Grounding	Mood regulation
Sedation		
Externalization	Emotional resolution	
Ventilation		

## Data Availability

The data are available on request from the corresponding author.

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
