# Peer review of "Online Song Intervention Program to Cope with Work Distress of Remote Dispatched Workers: Music for an Adaptive Environment in the Hyperconnected Era"

_behavsci, 2025, doi:10.3390/bs15070869_

Round 1

Reviewer 1 Report

Comments and Suggestions for Authors

Dear Authors,

I enjoyed reading and reviewing your manuscript on this important topic of using music within an interesting work setting that is not currently represented in the literature.  I would encourage you to specifically state that the approach is music therapy in the abstract and method, and provide further distinction on your education and training that qualify you for this work.  My concern is that people often think music cannot be harmful, and since it is a part of daily life, that anyone could facilitate this work, which could result in people not being appropriately trained to mitigate harm.

The revisions are most prevalent in the method section, as there are redundancies and sometimes conflicting information.  To decrease redundancy, I would suggest deleting the intervention section, as it overlaps with the procedure and program development sections.  Additionally, I would suggest that the participation format and data collection sections be combined, and make data analysis be a separate section.  Specific comments are indicated below, and my review is in favor of publication, pending revision.  Thank you for your work!

Abstract

  • Well written, identifying the theoretical model, method, and outcomes.
  • Please include background information as well. A summative statement related to the content in the first paragraph of the introduction would be well-positioned.

Literature Review

  • Please rework the purpose statement (p. 2, last paragraph) for improved clarity
  • Overall progression of content was presented well

Method—Please make sure to use the past tense consistently throughout the method

  • Participants:
    • How was “in good health” defined? (p. 3, participant inclusion criteria #3)
  • Intervention: This section can be deleted since it is redundant with subsequent method sections and has conflicting content. The comments below can be incorporated into the procedure.
    • “Designed based”—choose one. I would suggest based on.
    • What was included in the preliminary questionnaire?
    • Move the last sentence about group interventions before the therapist information.
  • Procedure:
    • The last sentence on page 4 states that music was provided through live performances or recorded tracks. This contradicts the intervention section, where it was reported to be online.  Please fix the appropriate section and check for consistent language.
    • For speakers and headphones, please include model information.
    • Group size is indicated as 5-7, which contradicts the intervention section.
    • In-take assessment—please clarify if it was one questionnaire or multiple (text and Table 2). Was it open-ended, rated, or both?
  • Program Development:
    • More detailed information is needed about the intervention. Please reference Robb, Carpenter, & Burns (2010) reporting guidelines for music-based interventions.
    • How many songs were included in the program? An appendix with a list of all songs would improve transparent reporting.
    • The frequency of sessions indicated here is different from the above intervention section. To reduce redundancy, move this to the participation format section.
    • Table 1 indicates that 2 participants had previous experience with therapy, which conflicts with this section.
    • “Working on lyrics” (paragraph before Table 3), please clarify. Are they doing lyric analysis, songwriting, or something else?
    • This section indicates that all sessions were provided online, but the procedure section indicated online and through live performances. Please clarify and provide specific information about how sessions were provided online, specifying the platform and who managed the setup on the ship.  To reduce redundancy, move this to the participation format section.
    • Go into more detail about the education and training for the interventionist and supervisor.
  • Participation Format:
    • Indicate sessions were recorded, where data was stored, and who had access. Were verbal processing skills used beyond the presented questions?
    • Indicate post-treatment questionnaires and what was included. Were the questionnaires orally reported?  If so, were these recorded too and then transcribed?
  • Data Collection & Analysis:
    • Where did the interview occur, and when within the treatment timeline did this occur?

Results

  • The first paragraph should be in the data analysis section instead of the results.
  • Table 5—Is the content in the parentheses participant statements? Please indicate what abbreviations stand for.
  • Paragraph between Tables 5 and 6—consistently use the past tense.

Discussion

  • Relate the discussion section to the review of literature instead of introducing new references.
  • Consistently use objective language instead of subjective (3rd paragraph—“certainly helped”, “very personal”)
  • Support findings statements with participants' statements or change verbiage to be more tentative.
  • Add a limitations section before future directions

References

  • References are formatted well, except for consistent use of capitalization. Please revise for consistency.

Author Response

Comments 1:

Abstract
• Well written, identifying the theoretical model, method, and outcomes.
• Please include background information as well. A summative statement related to the content in the first paragraph of the introduction would be well-positioned.

Response 1: Thank you for your suggestion. We have added one sentence of background information summarizing the first paragraph of the Introduction to the Abstract to enhance contextual clarity. (See Abstract, lines 1–3)

Comments 2:

 Literature Review
• Please rework the purpose statement (p. 2, last paragraph) for improved clarity
• Overall progression of content was presented well

Response 2: Thank you for your feedback. We have revised the statement of research purpose at the end of page 2 to make it clearer. (See Introduction page 3, lines 1–6)

Comments 3:

Method
• Please make sure to use the past tense consistently throughout the method

Response 3: Thank you for pointing this out. We have carefully checked the entire Methods section and revised the tenses to ensure consistency in using the past tense.

Comments 4: Participants
• How was “in good health” defined? (p. 3, participant inclusion criteria #3)

Response 4: Thank you for your question. We have revised this criterion to specify that participants were required to have no hearing impairments.

Comments 5: Intervention
• This section can be deleted since it is redundant with subsequent method sections and has conflicting content. The comments below can be incorporated into the procedure.
• “Designed based”—choose one. I would suggest “based on.”
• What was included in the preliminary questionnaire?
• Move the last sentence about group interventions before the therapist information

Response 5: Thank you for your detailed suggestions. We have removed the redundant content and integrated the relevant information into the “Procedure” section. The wording “designed based” has been revised to “based on.” Details of the preliminary questionnaire contents have been clarified. The sentence regarding group therapy has been relocated as suggested.

Comments 6:

Procedure
• The last sentence on page 4 states that music was provided through live performances or recorded tracks. This contradicts the intervention section, where it was reported to be online. Please fix the appropriate section and check for consistent language.
• For speakers and headphones, please include model information.
• Group size is indicated as 5–7, which contradicts the intervention section.
• In-take assessment—please clarify if it was one questionnaire or multiple (text and Table 2). Was it open-ended, rated, or both?

Response 6:

Thank you for your detailed feedback. We have standardized the terminology and revised the text to indicate that the intervention was delivered online. Regarding playback devices, participants used their own smartphones and headphones, and the specific models were unavailable; this has been clarified in the text. Group size has been consistently revised to 5–10 participants. The type of entry assessment questionnaires (open-ended and rating-based) has been specified. (See Methods, Procedure)

Program Development
• More detailed information is needed about the intervention. Please reference Robb, Carpenter, & Burns (2010) reporting guidelines for music-based interventions.
• How many songs were included in the program? An appendix with a list of all songs would improve transparent reporting.
• The frequency of sessions indicated here is different from the above intervention section. To reduce redundancy, move this to the participation format section.
• Table 1 indicates that 2 participants had previous experience with therapy, which conflicts with this section.
• “Working on lyrics” (paragraph before Table 3), please clarify. Are they doing lyric analysis, songwriting, or something else?
• This section indicates that all sessions were provided online, but the procedure section indicated online and through live performances. Please clarify and provide specific information about how sessions were provided online, specifying the platform and who managed the setup on the ship. To reduce redundancy, move this to the participation format section.

Response: Thank you for these essential recommendations. We have revised the section to comply with Robb, Carpenter, & Burns (2010) reporting standards. The full list of selected songs has been included in Appendix 2, while the song selection criteria have been provided in Appendix 1. Inconsistent information regarding intervention frequency has been removed. The discrepancy regarding participants’ prior therapy experience has been corrected. “Lyric creation” has been clarified as “lyric analysis.” The delivery format has been revised to consistently describe the intervention as online using Tencent video conferencing, with shipboard technical setup managed by an assigned officer. Details regarding the qualifications and backgrounds of the interventionist and supervisor have been fully described. (See Methods, Intervention Development & Appendix)

Participation Format
• Indicate sessions were recorded, where data was stored, and who had access. Were verbal processing skills used beyond the presented questions?
• Indicate post-treatment questionnaires and what was included. Were the questionnaires orally reported? If so, were these recorded too and then transcribed?

Response: Thank you for your attention to these important details. We have added descriptions indicating that all sessions were audio- and video-recorded with participants’ consent, securely stored on external hard drives accessible only to the principal therapist and research team. Post-intervention questionnaires were administered orally, with participants’ responses transcribed verbatim for analysis. (See Methods, Intervention Format)

Data Collection & Analysis
• Where did the interview occur, and when within the treatment timeline did this occur?

Response: Thank you for pointing this out. We have clarified that follow-up interviews were conducted online after the four-week intervention.

Results
• The first paragraph should be in the data analysis section instead of the results.
• Table 5—Is the content in the parentheses participant statements? Please indicate what abbreviations stand for.
• Paragraph between Tables 5 and 6—consistently use the past tense.

Response: Thank you for these helpful suggestions. We have removed redundant content and moved relevant parts to the “Data Analysis” section. Clarification regarding the meaning of participant codes in parentheses has been provided. The Results section has been revised to consistently use the past tense.

Discussion
• Relate the discussion section to the review of literature instead of introducing new references.
• Consistently use objective language instead of subjective (3rd paragraph—“certainly helped”, “very personal”)
• Support findings statements with participants' statements or change verbiage to be more tentative.
• Add a limitations section before future directions

Response: Thank you for your constructive feedback. We have revised the discussion to more closely connect with the reviewed literature. Subjective wording has been replaced with objective language. Quotations from participants have been added where appropriate to support conclusions. Additionally, we have included a “Study Limitations” subsection before the “Future Directions” section.

References
• References are formatted well, except for consistent use of capitalization. Please revise for consistency.

Response: Thank you for your careful review. We have checked all references to ensure consistency in capitalization and formatting.

4. Response to Comments on the Quality of English Language

Point 1: The English could be improved to more clearly express the research.

Response 1:  Thank you for pointing this out. We have thoroughly reviewed the manuscript and revised verb tenses throughout to maintain consistency.

5. Additional clarifications

We would like to emphasize our gratitude for the reviewer’s thoughtful comments, which have substantially contributed to improving the manuscript. All suggested revisions have been incorporated, and all changes are clearly marked in the tracked version of the manuscript.

Reviewer 2 Report

Comments and Suggestions for Authors

This manuscript addresses an important issue -the mental well-being of remotely dispatched workers- through an innovative online song-based intervention. The 4-week program, grounded in the contextual support model, is thoughtfully designed and provides rich qualitative data that contribute to our understanding of emotional regulation, social connection, and autonomy in isolated work settings. The clarity of presentation and practical applicability, especially regarding online scalability, enhance the value of the study.  However, I believe the manuscript would benefit from further refinement and clarification in several areas. Please refer to the attached file for detailed comments and suggestions.  

Comments on the Quality of English Language

The overall use of English is adequate: however, a thorough language review is recommended to improve clarity, coherence, and academic tone throughout the manuscript.  

Author Response

Comments 1: While the study presents a well-designed intervention and yields meaningful qualitative findings, it is important to address the gender homogeneity of the participant group...

Response 1: Thank you for pointing out this important issue. We fully acknowledge this limitation. A specific statement has been added to the Discussion section (p. 15, paragraph 2, lines 3–7), addressing the lack of gender diversity among participants and its potential influence on the generalizability of the findings. Additionally, we have suggested that future studies recruit a more diverse participant group to enhance the applicability of the results.

Comments 2: The manuscript states that individual sessions used personally meaningful songs, while group sessions involved 'frequently selected songs'...

Response 2: Thank you for this valuable suggestion. We have added a clarification explaining that their selection was based on preference surveys and frequency data collected during the pre-intervention assessments, combined with thematic relevance to autonomy, relatedness, and competence (p. 6, paragraph 2, lines 4–8).

Comments 3: On page 6, the manuscript explains that participants identified 'positive personal resources' as part of the four-week plan...

Response 3: Thank you for your insightful comment. The section has been revised to define 'positive personal resources,' specifying emotional strengths, coping skills, past meaningful experiences, and social support systems (p. 6, paragraph 3, lines 2–7).

Comments 4: While the manuscript indicates that the program was conducted entirely online with music played virtually, more detail is needed on the delivery format...

Response 4: Thank you for raising this point. We clarified that all sessions were delivered via video conferencing (Tencent Meeting, enterprise version), with real-time video used to facilitate participant engagement (p. 4, paragraph 3, lines 4–8).

Comments 5: Additional technical details would strengthen the manuscript’s methodological transparency...

Response 5: Thank you for this helpful suggestion. Sessions were conducted when the ship was anchored in port to ensure stable internet. Repeated audio and signal tests were performed to optimize communication stability (p. 4, paragraph 3, lines 8–14).

Comments 6: While the study offers valuable insights, the authors may wish to reflect on how gender may influence participants’ emotional engagement...

Response 6: Thank you for this meaningful suggestion. We incorporated this into the Discussion (p. 15, paragraph 2, lines 7–12), highlighting the potential moderating role of gender.

Round 2

Reviewer 2 Report

Comments and Suggestions for Authors

The authors have responded thoroughly and accurately to comments regarding the 6 key issues raised. Each revision was appropriately addressed, and the manuscript has been substantially improved.  One minor point (p.18): it many be preferable to correct the double parentheses in the following expression- ((IRB No: FZU-PSY 2024-0165)